# Indium Tin Oxide Thin-Film Thermocouple Probe Based on Sapphire Microrod

**DOI:** 10.3390/s20051289

**Published:** 2020-02-27

**Authors:** Jinjun Deng, Linwei Zhang, Liuan Hui, Xinhang Jin, Binghe Ma

**Affiliations:** Key Laboratory of Micro/Nano Systems for Aerospace, Ministry of Education, Northwestern Polytechnical University, Xi’an 710072, China; zhanglw@mail.nwpu.edu.cn (L.Z.); huiliuan@mail.nwpu.edu.cn (L.H.); jxh@mail.nwpu.edu.cn (X.J.); mabh@nwpu.edu.cn (B.M.)

**Keywords:** thin-film thermocouple, sapphire cylinder, indium tin oxide, sputtering

## Abstract

Indium tin oxide (ITO) thin-film thermocouples monitor the temperature of hot section components in gas turbines. As an in situ measuring technology, the main challenge of a thin-film thermocouple is its installation on complex geometric surfaces. In this study, an ITO thin-film thermocouple probe based on a sapphire microrod was used to access narrow areas. The performance of the probe, i.e., the thermoelectricity and stability, was analyzed. This novel sensor resolves the installation difficulties of thin-film devices.

## 1. Introduction

Indium tin oxide (ITO) thin-film thermocouples are promising temperature sensors for gas turbines (even in harsh environments). Thermoelectric films offer fast response because of their submicrometer thickness. Therefore, they do not affect the flow field state [1,2,3,4].

For in situ measurements, the main challenge is the installation of thin-film thermocouples on complex geometric surfaces. Thin-film thermocouples consist of two legs as thin films, which are often directly deposited on the measuring surface. A key problem is to form a particular shape of thin-film legs on curved surfaces. It is difficult to place such components in a deposition device. Additionally, the in situ lead connection has the disadvantage that the device alters the surface of the object of interest and may negatively interfere with the surrounding flow field as well as increase the risk of connection failure because of the rapid flow scour [5,6,7].

Another limitation is that there is no compensation wire that matches a ceramic thermocouple, such as the ITO/In_2_O_3_ thermocouple. Our research team previously developed and studied an ITO film thermocouple (In_2_O_3_:ITO 90:10) using hole leads to monitor the temperature gradient of a turbine guide blade [8]. The thermocouple reduced the influence of the conductor on the flow field to some extent, but the measured temperature gradient was not large because of the deposition of thin films on the turbine blades with confined dimensions. Lead wires cannot be used to extend the cold port. 

In this study, a novel thin-film thermocouple probe is introduced to solve installation difficulties for thin-film devices. The thermal probe is deposited on an insulation rod of minuscule diameter, which makes it easy to monitor the temperature gradient in narrow areas. Specifically, a hot junction of ITO/In_2_O_3_ thermocouple was prepared on one head of the sapphire microrod, and two thermal legs were directly prepared on the microrod. The performance of the thin-film thermocouples fabricated directly on a sapphire substrate was analyzed at 20–871.26 °C.

Such thin-film thermocouple probes have several advantages. As shown in Figure 1, this probe makes it easy to monitor the temperature of the edge plate on the turbine guide blade. A probe head with hot junction should be flush with the upper surface of the edge plate through an additional prepared measuring hole. In this way, in situ film preparation is converted on a complex surface to a regular cylindrical surface, and it is only the hot-junction thin films that affect the flow field and not the lead nodes. 

## 2. Sensor Design and Fabrication

### 2.1. Material Selection and Sensor Structure

Figure 2 shows the ITO thin-film thermocouple probe and the fabricated sensor. The sensor comprised a sapphire microrod and a corundum tube. Microrods served as rigid support for the sensitive films, and the corundum tube protected the lead nodes. In this study, ITO (In_2_O_3_:SnO_2_ 90:10 by wt %) and In_2_O_3_ were employed as thermoelement materials. In [9], it was shown that thin-film thermocouples based on In_2_O_3_ and ITO exhibited higher sensitivity than metallic thin-film thermocouples, and the Seebeck coefficient of In_2_O_3_/ITO thin-film thermocouples was 170 μV/°C.

The sensor body comprised a short section of the sapphire microrod with a diameter of 1 mm. The melting point of sapphire can reach 2050 ℃, and its thermal expansion coefficient (8.4 × 10^−6^/K) is close to that of ITO (7.26 × 10^−6^/K), which led to low thermal expansion stress. Moreover, the head of the rod had a beveled edge (Figure 3). This chamfered structure is beneficial for film stability at high temperatures. Two sensing films were directly deposited on the microrod as thermocouple legs and connected to the end face as a hot junction. The projection width of the thermocouple legs was approximately 0.2 mm. The sensor body, which included the film structure and the chamfer structure, is shown in Figure 3.

Ag paste was used to bond the Pt wires to the thermocouple legs. Pt wires with a diameter of 0.1 mm were selected to transmit the thermoelectrical power to ensure the stability of the wire resistance. A corundum tube (inner diameter of 5 mm) filled with ceramic adhesive (919 powder) was used to enhance the mechanical strength of the connection, and a commercial K-type thermocouple (wire) was embedded to measure the cold-junction temperature of the thin-film thermocouple probe. Moreover, another tube (outer diameter of 2 mm) deposited with sensing films was used to connect the microrod to Pt wires and was employed as an extension wire, avoiding a connection between the two sensing films due to the Ag paste coating on the cold junction.

### 2.2. Film Preparation

A thin-film thermocouple comprises two electrodes, and surface graphics must be considered during the sputtering process. In this study, we used a graphical preparation method for thin films deposited on curved surfaces. 

The brief steps of the film preparation process are shown in Figure 4. The sapphire cylinder was first clamped on a supporting plate with a ceramic fixture, and polyimide (PI) tape was then attached to the surface of the fixture–cylinder assembly (Figure 4a,b). The fixture was a square ceramic plate with a groove corresponding to the diameter of the sapphire cylinder, and the thickness of the fixture was equal to the diameter of the cylinder. High-accuracy laser marking was performed to etch rectangular patterns of electrodes on the PI film as a soft mask (Figure 4c). Then, the sensing film was sputtered on the PI mask, and the mask was subsequently removed mechanically (Figure 4d,e). One thermocouple leg was prepared. The other leg was developed similarly, except that the cylinder had to rotate (Figure 4f). 

The main challenge in the graphical process was the alignment between the etching pattern on the PI mask and the fixture groove. The support plate was attached to the worktable of the laser marking machine with a positioning fixture, and the groove trace was etched on the surface of the support plate. Then, the support plate was removed and attached to the ceramic fixture. The trace was aligned with the groove, and the support plate/fixture was secured with PI tape as a composite fixture. Finally, the sapphire cylinder was clamped with the composite fixture.

Radiofrequency (RF) magnetron sputtering was selected as the deposition method for fabricating the sensing films on the sapphire cylinder. It is widely used because of its features such as low temperature, uniform film deposition, and process controllability. RF magnetron sputtering is particularly useful for preparing insulating films [10]. An MSP-3470x sputtering apparatus was used with the process parameters in Table 1. ITO (In_2_O_3_:SnO_2_ 90:10 by wt %) and In_2_O_3_ were used as target materials. Presputtering was performed for 30 min to obtain a clean and rough surface, reinforcing the film–substrate interface [11]. The sputtering was performed for 30 min to form two thermocouple legs. The film thickness at the hot junction was estimated to be 1.5 μm according to the deposition rate.

Next, the sensing films were subjected to thermal annealing. Figure 5 shows scanning electron microscopy (SEM) images of the ITO films before (Figure 4a,b) and after (Figure 5c,d) the thermal annealing. The annealing temperature was 1000 ℃, and the holding time was 4.5 h. Figure 4a indicates that the thin film deposited via RF magnetron sputtering was compact and uniform. However, the film had microcracks (Figure 5b). Figure 5c shows the influence of thermal annealing on the morphology of the film surface. Some microstructures of the film changed into cellular crystals, and the particle boundaries became apparent. The thin film did not transform completely (Figure 5d).

## 3. Results and Discussion

The functional principle of thermocouples is based on the Seebeck effect, which describes the thermodiffusion of charge carriers as a function of the temperature [12]. A thermocouple consists of two different metals connected to the hot tip (*T_h_*) and the cold point (*T_c_*). The measured thermodiffusion voltage depends on the difference between *T_h_* and *T_c_*, as well as the Seebeck coefficients (S) of the metals. In this study, the Seebeck coefficient was given as
(1)$$S=\frac{\Delta V}{\Delta T}=\frac{\Delta V}{T_{h}-T_{c}}$$
where *S* represents the Seebeck coefficient (μV/°C), Δ*V* represents the potential difference between the two thermocouple materials, and Δ*T* represents the temperature difference between the hot junction (*T_h_*) and the cold junction (*T_c_*). For calibration, another K-type thermocouple was placed close to the sensing surface of the thermocouple probe to measure *T_h_*. The thermal voltages of the ITO/In_2_O_3_ thermocouple probe and the K-type thermocouples were recorded using a digital multimeter (Keithley K2002) controlled by LabVIEW. The calibration system is shown in Figure 6.

The thermoelectric characteristic output of the sensor is shown in Figure 7. To obtain accurate values, the temperature measurement range, temperature interval, and holding time were set as 20–900 ℃, 100 ℃, and 90 min, respectively. As shown in Figure 7a, the thermocouple voltage of the calibrated sensor agreed with that of the K-type thermocouple. The peak temperature of the hot junction reached 871.26 ℃. Obviously, the signal of the thin-film thermocouple probe contained a large amount of noise at a high temperature (>766 ℃), while the K-type thermocouple consistently remained stable. According to logical analysis, an uneven film structure makes the output of the sensor unstable at high temperatures.

The fitted static characteristic curve is shown in Figure 7b. It can be described as follows:(2)$$\begin{array}{ll} V(\Delta T) & =-2.801\times 10^{-7}\times \Delta T^{3}+2.985\times 10^{-4}\times \Delta T^{2}\\ & \;\;+0.04151\times \Delta T+0.7202 \end{array}$$
where Δ*T* (℃) represents the temperature difference between *T_h_* and *T_c_*, and *V*(Δ*T*) represents the thermocouple voltage (mV). When the temperature reached the peak value of 871.26 ℃, the temperature difference was 652.72 ℃, the thermocouple voltage was 77.04 mV, and the Seebeck coefficient was 118.03 μV/℃. The static curve exhibited a slight deviation in linearity, i.e., a stretched S-shaped akin to a logistic function, which was attributed to phonon–electron interactions at different temperatures [13]. 

The drift in the Seebeck coefficient during the insulation state is shown in Figure 8. Data containing gross errors (exceptional data) were ignored. To eliminate the influence of temperature fluctuations, the Seebeck coefficient was used instead of the thermal voltage to evaluate the stability error. The temperatures of the hot and cold junctions in the insulation state are shown in Figure 8. An exponential model was adopted to describe the relationship between the Seebeck coefficient and the holding time, as follows:(3)$$S=2.948e^{-8.723t}+117.9$$
where *S* represents the Seebeck coefficient (μV/°C), and *t* represents the holding time (h). This equation is only used to describe the numerical variations of the Seebeck coefficient; it is not of great significance to the present study. The drift of the Seebeck coefficient stability decreased by 2.39% over the 1.4 h test period, and the drift amount decreased by 2.35% within 0.5 h. The temperature difference measured by the K-type thermocouples had a precise lag compared with the actual temperature difference between the hot and cold junctions. Initially, the measured temperature was lower than the actual temperature, which led to a higher Seebeck coefficient.

## 4. Conclusions

An ITO thin-film thermocouple probe was fabricated using a novel method, and the thermoelectric output of the sensor in the range of 20–871 ℃ was investigated. Additionally, a method to prepare a curved film using laser marking was developed. According to a static test, the temperature probe had an average Seebeck coefficient of 124.50 μV/℃, and the drift decreased by 2.39% within 1.4 h at high temperatures (maximum 871 ℃). This novel device can be used to measure the temperature in narrow areas, not just in the marginal plate of a turbine blade. Future research will focus on improving the signal-to-noise ratio of the thermoelectric output, considering the effect of thermal annealing on the thermoelectric output.

## Figures and Tables

**Figure 1 sensors-20-01289-f001:**
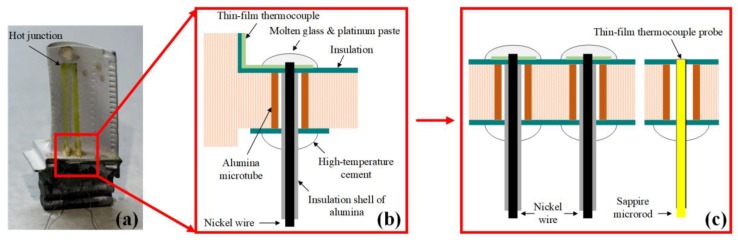
(**a**) Turbine blade with indium tin oxide (ITO) thin-film thermocouple, (**b**) structural diagram of the turbine blade through-hole lead connection, and (**c**) thin-film thermocouple probe used to measure the surface temperature.

**Figure 2 sensors-20-01289-f002:**
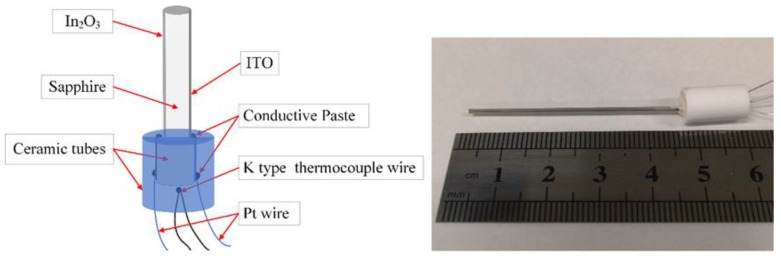
(**a**) Schematic of the ITO thin-film thermocouple probe and (**b**) the fabricated sensor.

**Figure 3 sensors-20-01289-f003:**
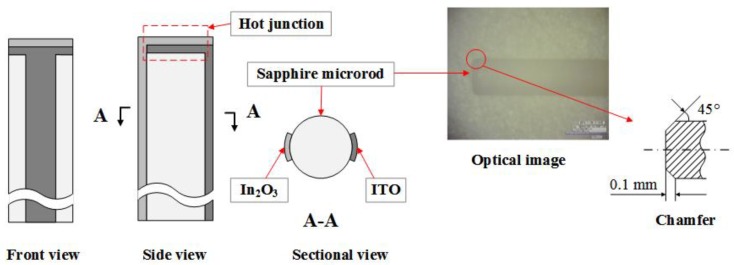
Structure of the sensor.

**Figure 4 sensors-20-01289-f004:**
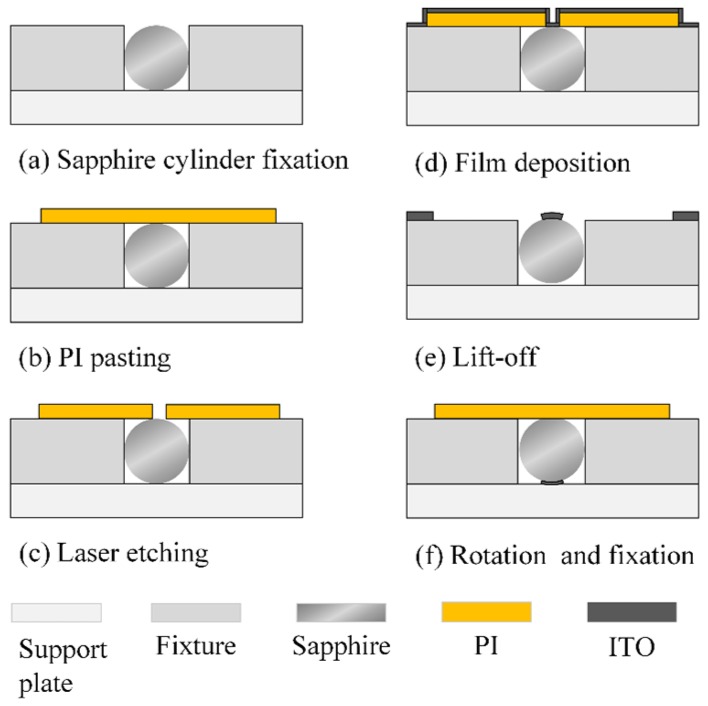
Graphical processing of curved films.

**Figure 5 sensors-20-01289-f005:**
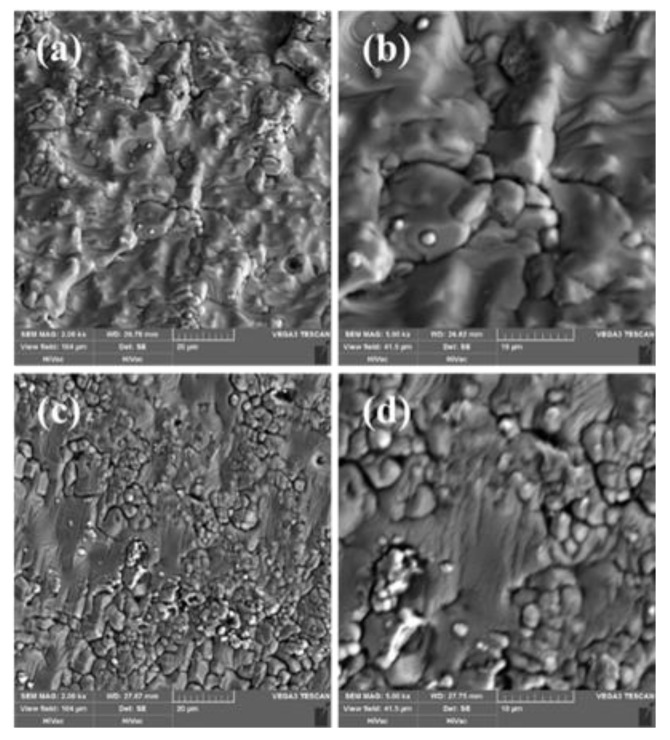
SEM images of ITO films deposited on the sapphire microrod before (**a**,**b**) and after (**c**,**d**) thermal annealing at 1000 °C.

**Figure 6 sensors-20-01289-f006:**
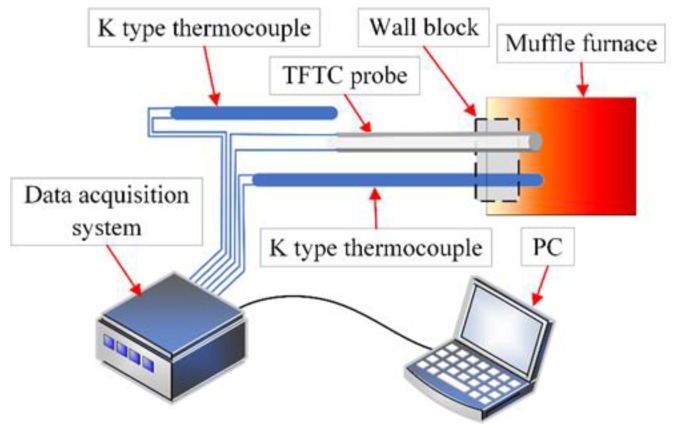
Calibration system of the thin-film thermocouple probe.

**Figure 7 sensors-20-01289-f007:**
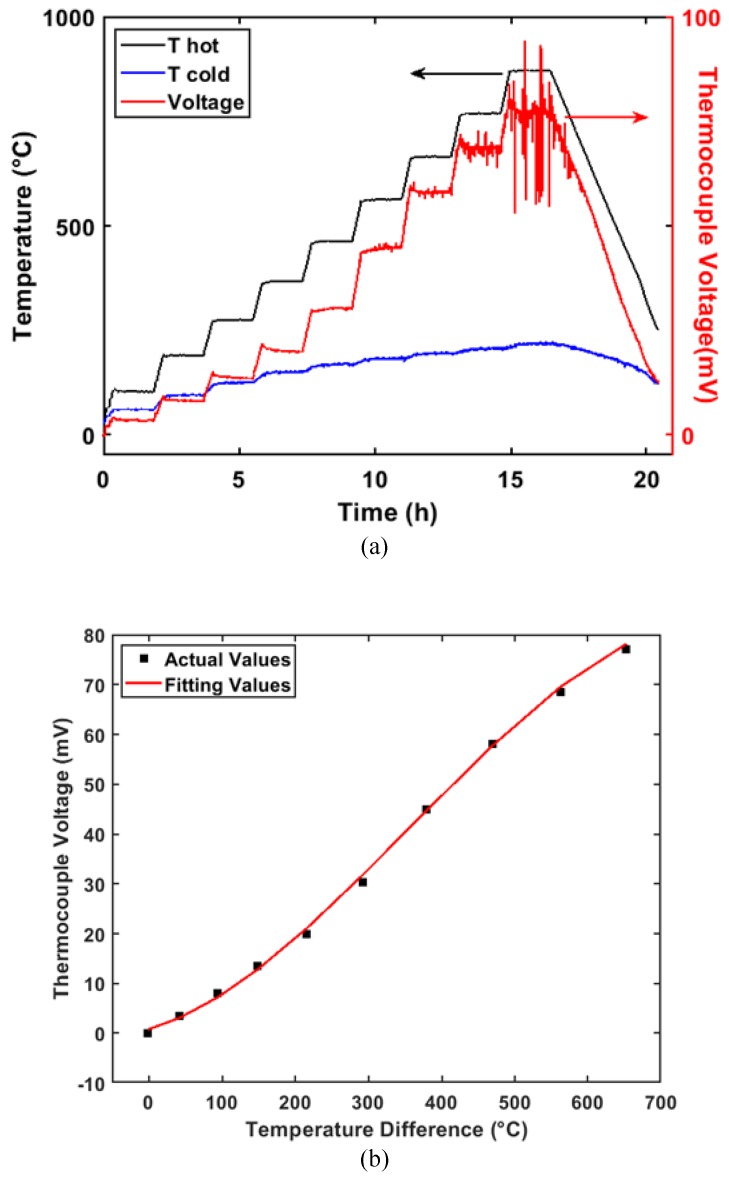
(**a**) Thermoelectric output of the In_2_O_3_:ITO 90:10 thin-film thermocouple probe. The peak temperature of the hot junction was 871.26 ℃. (**b**) Fitted static characteristic curve. The average Seebeck coefficient was 124.50 μV/℃.

**Figure 8 sensors-20-01289-f008:**
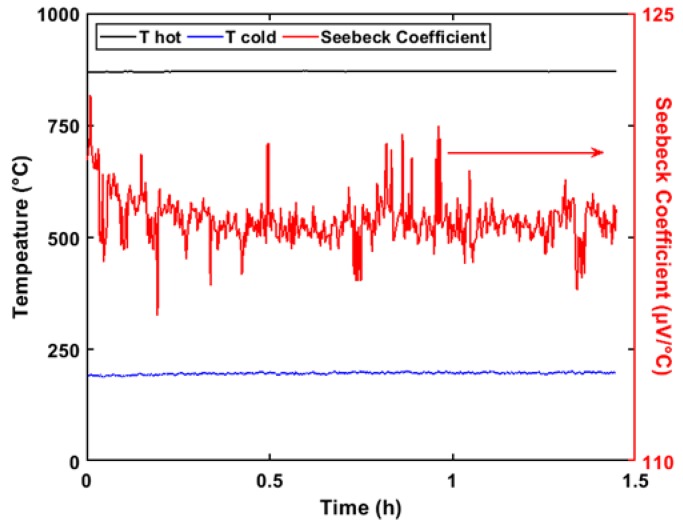
Stability error of the thin-film thermocouple probe. The Seebeck coefficient decreased from 120.85 to 117.96 μV/°C in 1.4 h.

**Table 1 sensors-20-01289-t001:** Sputtering values used for the deposition of thin-film thermocouple legs and junctions.

Parameters	Values
Base pressure	8 × 10^−4^ Pa
Ar flow rate	120 sccm
Sputtering gas pressure	1.5 Pa
Target power	200 W
Deposition rate	23.0 nm/min (In_2_O_3_) and 27.1 nm/min (ITO)

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
