# Peer review of "Indium Tin Oxide Thin-Film Thermocouple Probe Based on Sapphire Microrod"

_sensors, 2020, doi:10.3390/s20051289_

Round 1

Reviewer 1 Report

The authors presented a method to fabricate ITO/In2O3 thin-film thermocouple on a thin, long rod, which had a curved surface. The device showed an average sensitivity around 120 μV/℃ in the testing range from room temperature to 870 ℃. While when the testing temperature was above 750 ℃, output of the device was quite nosy. The device was well calibrated and the performance of the probe in terms of sensitivity and stability was analyzed. It may be applied to monitor temperature of hot section components in gas turbines and other fields with complex geometric shape.

A few minor points:

1, There are some typos, e.g., in the introduction section (line22), "ssub-micrometer".

2, In Figure 2, there is no scale bar in the optical image, and the image is not clear.

Author Response

Dear Reviewer:

    Thank you for reviewing and pointing out my paper. I would like to give you the following reply for your Suggestions.

1, There are some typos, e.g., in the introduction section (line22), "ssub-micrometer".

   I am so sorry for such an editing error and I have modified the original text.

2, In Figure 2, there is no scale bar in the optical image, and the image is not clear.

I found the original image which attachs the scale.I have adjusted the sharpness of the image.Please check it.

Reviewer 2 Report

General remarks:

Grammar must be greatly improved in the Abstract, Introduction and Sensor design and fabrication parts.

Introduction is not clear, must be greatly improved to show novelty of actual design. How does it overcome mentioned issues in the introduction? This is, 1. Difficulty of installation, 2. Thin-film shape on curved surfaces, 3. Interaction of probe with the surrounding flow field.

Specific remarks:

line 10: substitute the word technique for device

line 14: "to access narrow areas"

line 49-50: Average Seebeck coefficient of which material? Please add reference for the value of 170 uV/ºC

lines 135-138: How is the noise being generated? Can it be produced by interconnection (silver paste) detachment? Can you prove this somehow?

line 142: Instead of temperature gradient, it should be temperature difference.

lines 146-147: Seems unnecessary information to me.

lines 155-168: Please clarify: Is the drift produced by a material change or is it spurious data coming from the K-type thermocouple lag? If it is because of a material change, how can you explain this change at ~871ºC when the sample has been previously annealed at 1000ºC?

Figure 7 is labeled as figure 6

Reviewer 3 Report

The reviewed manuscript attempted to fabricate a 1.5 μm thin ITO film thermocouple on a sapphire cylinder and investigate the thermoelectricity and stability properties. The authors claimed that the Seebeck coefficient was 124.50 μV/oC and the drift amount decreased by 2.39% within 1.4h at high temperature of 871oC.

I cannot recommend publication of this manuscript, which is not innovative. There is also no in-depth mechanism analysis. Besides, there are many flaws in this manuscript:

1.English expression needed to be improved;

2.The authors didn’t say the effect of annealing on the properties of ITO thin film;

3.The authors didn’t give the response time of ITO thin film during thermoelectric output test;

4. Fig.7 didn’t show in this article;

5.Temperature cycling experiment at high temperature is needed to show the repeatability of the sensor device.

Author Response

Dear Reviewer:

    Thank you for reviewing and pointing out my paper. I would like to give you the following reply for your suggestions.

English expression needed to be improved;

I am so sorry for such language error and I have modified the original text.

The authors didn’t say the effect of annealing on the properties of ITO thin film;

Thermal annealing of the film after sputtering can effectively reduce film defects and lattice dislocations. With the input of external energy, the lattice defects of point defects in the film were recombined. Thin films, on the other hand, have a recrystallization effect. As the temperature increases, the columnar crystals gradually grow into cell-like crystals close to the bulk material, and the high-temperature resistance and thermoelectric stability of the film are improved.

The main application scenario of the probe-type thin-film thermocouple sensor is the surface of the tongue and groove of the turbine blade, whose temperature does not exceed 1000 ℃. Therefore, the thermal annealing temperature of this paper is selected to be 1000 ℃, and the single insulation time is initially set to 1.5h. A total of three annealings are performed Insulate, then record the output voltage of the sensor at the same temperature. The test data results are shown in the following figure.

Effect of thermal annealing time on the thermoelectric characteristics of thin film thermocouples

non-linear error

Average Seebeck coefficient

Heating Cycle 1

27.87%

88.6μV/℃

Heating Cycle 2

0.64%

105.2μV/℃

Heating Cycle 3

3.87%

115.6μV/℃

It can be seen that as the holding time increases, the average Seebeck coefficient of the thin-film thermocouple will gradually increase, and thermal annealing can effectively reduce the non-linear error of the thin-film thermocouple output. However, with the increase of the holding time, the output non-linear error increases again.

Comprehensive consideration, the heat treatment of the thin-film thermocouple sensor should be selected to 1000 ℃, holding for 3 to 4 hours.

The authors didn’t give the response time of ITO thin film during thermoelectric output test;

The response time of ITO film was not tested, but the response time of the thermocouple device was tested. The constant temperature contact thermal excitation method is used to test the dynamic test. The heat source is molten solder and the excitation load time is on the order of milliseconds. This method is used to initially evaluate the dynamic response performance of the sensor. Test results show that the device's thermal response time is 1.513ms.

Drop molten tin liquid test sensor dynamic response characteristics

7 didn’t show in this article;

Figure 7 is labeled as figure 6. I am so sorry for such edit error and I have modified the original text.

Temperature cycling experiment at high temperature is needed to show the repeatability of the sensor device.

Two repeatability tests were carried out on the device, and the results showed a good consistency.

Temperature cycle test results

Round 2

Reviewer 2 Report

In this 2nd version I have just 2 minor comments

It would be nice to have an image / scheme of the turbine with the previous and new thermocouples used to measure temperature. Equation 3 has both a negative and a positive exponential terms. The positive one has a very small exponent coefficient. Wouldn't it be better to just define that coefficient as 0 and thus substitute the whole exponential by just 1? A positive exponential means that the Seebeck coefficient de-stabilizes with time, which doesn't make much sense unless there are material changes that are not discussed in the manuscript.

Author Response

Dear Reviewer:     Thank you for reviewing and pointing out my paper. I would like to give you the following reply for your suggestions.
